# Chirality detection of enantiomers using twisted optical metamaterials

Yang Zhao[1], Amir N. Askarpour[1,2], Liuyang Sun[3], Jinwei Shi[4], Xiaoqin Li[3] & Andrea Alù[1]

Many naturally occurring biomolecules, such as amino acids, sugars and nucleotides, are inherently chiral. Enantiomers, a pair of chiral isomers with opposite handedness, often exhibit similar physical and chemical properties due to their identical functional groups and composition, yet show different toxicity to cells. Detecting enantiomers in small quantities has an essential role in drug development to eliminate their unwanted side effects. Here we exploit strong chiral interactions with plasmonic metamaterials with specifically designed optical response to sense chiral molecules down to zeptomole levels, several orders of magnitude smaller than what is typically detectable with conventional circular dichroism spectroscopy. In particular, the measured spectra reveal opposite signs in the spectral regime directly associated with different chiral responses, providing a way to univocally assess molecular chirality. Our work introduces an ultrathin, planarized nanophotonic interface to sense chiral molecules with inherently weak circular dichroism at visible and near-infrared frequencies.

[1] Department of Electrical and Computer Engineering, The University of Texas at Austin, 1 University Station, Austin, Texas 78712, USA. [2] Department of Electrical Engineering, Amirkabir University of Technology, 424 Hafez Avenue, Tehran 15875-4413, Iran. [3] Department of Physics, The University of Texas at Austin, 1 University Station, Austin, Texas 78712, USA. [4] Department of Physics, Beijing Normal University, 19 Xinjiekou Outer Street, Beijing 100875, China. Correspondence and requests for materials should be addressed to A.A. (email:alu@mail.utexas.edu).

The handedness of enantiomers is strongly associated with their pharmacological effects, especially their potency and toxicity[1]: single enantiomer drugs are often more efficient than their racemic mixtures[2]; more importantly, while one chirality forms a powerful medicament, the other one may cause serious side effects[3,4]. To date, molecular chirality is routinely determined using circular dichroism (CD) measurements[5,6]. However, chiral molecules on their own typically possess extremely small CD, in the range of only few tens of millidegrees observable in the ultraviolet range[6]. Consequently, conventional CD measurements require up to 30 min of integration time to resolve these small signals, and they involve sub-millilitre volumes at a molecular concentration in the micromolar range[6]. These measurements need to be carried out during and after each synthetic step of a chiral compound, which makes it particularly challenging at the beginning of a synthetic approach development when large amounts of analytes are difficult to obtain due to limited production efficiency. Therefore, the capability of detecting the handedness of enantiomers with significantly less concentration in millisecond time scales would be extremely beneficial for pharmaceutical applications.

Recent works have discussed the use of plasmonic and dielectric nanostructures to enhance weak CD effects for chiral molecule detection. Chiral metasurfaces[7–11] and achiral plasmonic[12–15] and silicon nanospheres[16] have been explored in this context. The CD response of a chiral molecule may be significantly boosted around the resonance frequency of a plasmonic nanosphere[12] (Fig. 1b), with a resultant CD still in the millidegree range, but now extended to the visible spectrum. This spectral extension is advantageous because long exposure to ultraviolet radiation can severely affect biomolecular samples. However, plasmonic achiral structures do not allow a univocal chiral detection because the sign of the CD spectrum is sensitive to the binding orientation of the chiral molecule to the plasmonic nanoparticles[15]. Chiral metasufaces[7–10], in principle, allow unambiguous detection of large chiral molecules. The first chiral detection using a metasurface was reported in ref. 7, in which a single layer gammadion structure was used for detecting various proteins. The main mechanism in this work relies on measuring the spectral shift in the far field spectrum due to the nearfield interactions between the chiral molecules and the metasurface. The same mechanism was later extended to different chiral molecules[8–11]. However, further improving the detection sensitivity based on spectral shift is limited, because the final measured spectrum is typically combined with the background CD spectrum of the metasurface. It is known that introducing molecules in the near-field of plasmonic structures can also result in spectral shift due to refractive index changes, regardless of their handedness, which makes a direct detection of molecular chirality difficult for weak signals coming from low concentrations of molecules.

In this paper, we introduce a platform for chirality detection based on a plasmonic 'twisted' metamaterial that overcomes the above-mentioned limitations. The near-field of these meta-materials can be optimally tailored to amplify chiral–chiral interactions between the metamaterial inclusions and the molecules, enabling high-sensitivity handedness detection of enantiomers (schematically shown in Fig. 1c). We observe an enhanced cumulative CD spectrum of molecules reaching up to a few degrees, two orders of magnitude larger than what conventional CD spectroscopy can typically measure[17]. Our design provides additional flexibility to easily tune the enhanced CD of the molecules to a desired frequency, where the solvent may exhibit minimal absorption. In addition, our approach is capable of detecting the enhanced CD response of the molecules isolated from the large background CD spectrum of the metamaterial, without relying on any spectral shift. Our proposed method, after proper post-processing that makes it robust against fabrication imperfections, allows us to reliably separate the responses associated with opposite CD signs from enantiomers in as low as 55 zeptomoles, an amount that is $10^{15}$ times less than what typical commercial CD spectroscopy tools are able to detect.

## Results

**Twisted metamaterials to detect chiral molecules.** The CD response of a chiral sample is defined as the difference in transmitted power between right- and left-handed circularly polarized light, normalized to the total transmitted power for the two circularly polarized excitations. By adsorbing a monolayer of chiral molecules on a metamaterial surface and assuming a chirality $\kappa_m$ from the chiral molecules near the CD resonance of the metamaterial, the output CD response, $CD_o$, of the chiral molecule-metamaterial assembly can be written as (see Supplementary Notes 1 and 2)

$$CD_o = CD_i + 4kw \, \mathrm{Im}[\kappa_m] \frac{|T_{LR}T_{RL}|^2 - |T_{LL}T_{RR}|^2}{\left(|T_{LR}|^2 + |T_{RR}|^2\right)^2 + \left(|T_{LL}|^2 + |T_{RL}|^2\right)^2},$$

$$(1)$$

where $CD_i$ denotes the background CD of the metamaterial without chiral molecules; $T_{LR}$ is the left-handed circularly polarized transmission coefficient of the metamaterial with a right-handed circularly polarized input (same nomenclature for the other $T_{ij}$ coefficients); $k = 2\pi/\lambda$ is the wave vector with $\lambda$ being the effective wavelength in free-space, and $w$ is the thickness of the molecular monolayer, for which we assume $kw \ll 1$.

It is seen that the output CD of the loaded metamaterial is controlled by two relevant terms: the background CD of the metamaterial ($CD_i$) and the imaginary part of the chirality coefficient $\kappa_m$, which corresponds to the CD of the molecule near the CD peak frequency of the metamaterial. As we show in the following, both terms may be properly enhanced using twisted metamaterials[18,19], formed by stacking two or more closely spaced achiral metasurfaces with a proper sequential rotation between neighbouring layers. This geometry is ideally suited for our purpose: first, it is based on a planarized surface over which it is easy to deposit molecules with a controllable density; second, it can be composed of simple achiral inclusions (in our geometry

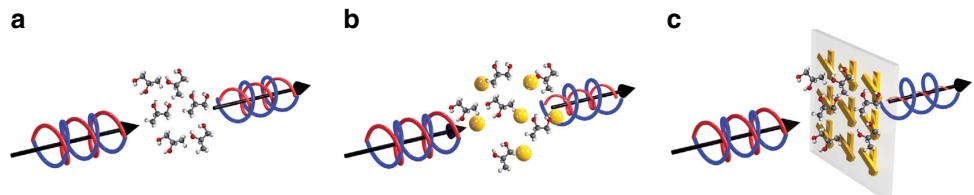

**Figure 1 | Schematic plot of enhanced chiral sensing.** (**a**) Circularly polarized light impinges onto assemblies of chiral molecules. It normally induces small CD in the ultraviolet region, in the millidegree range. (**b**) Chiral sensing with achiral plasmonic nanostructures shifts the CD peak to the resonance of the plasmonic nanoparticles. (**c**) Enhanced chiral sensing with twisted plasmonic metamaterials, resulting in orders of magnitude enhancement of CD signals.

gold nanorods) that can significantly boost local light-molecule interactions; most importantly, this geometry retains a giant chiral response associated with their rotation in the stacking direction[19–30]. To demonstrate these properties and identify the optimal geometry, we constructed several twisted metamaterials composed of two stacked metasurfaces separated by an 80 nm dielectric layer with densely packed nanorod inclusions and five different twisting angles from one metasurface to the second, spanning from $\pm 30°$ to $\pm 90°$ at an interval of $\pm 15°$. By optimizing the relative twist angle between the two layers, we select both peak wavelength and maximum value of $CD_i$, maintaining a high fidelity of fabrication (see Supplementary Note 3).

**Field enhancement versus chirality enhancement**. The optimized twisted metamaterial geometry retains another peculiar property that allows maximizing the chirality detection of the adsorbed molecules, associated with efficiently boosting the coefficient $\kappa_m$ in equation (1). As we discuss in the following, although $\kappa_m$ is closely related to the intrinsic chirality of the molecules, its effective value in equation (1) can be enhanced by suitably engineering the near-field light-molecule interaction sustained by the plasmonic particles forming the metamaterial. The effective chirality $\kappa_m$ can be boosted by two distinct mechanisms, based on Poynting's theorem and non-reciprocity theorem. The first mechanism is based on the near-field enhancement factor $\mathscr{F}$, defined as the near-field intensity in the vicinity of the metamaterial surface normalized to the far-field intensity $(\mathscr{F}=|\mathbf{E}_{\text{Nearfield}}|^2/|\mathbf{E}_{\text{Farfield}}|^2)$. This factor is associated with a larger local density of states supported by achiral plasmonic effects, consistent with the phenomenon schematically sketched in Fig. 1b and considered in recent papers[12,15]. Although this effect is normally important for refractive index plasmonic sensing, it is not beneficial for the proposed scheme of chirality detection. More interestingly, the effective molecular chirality $\kappa_m$ can be enhanced by the chiral enhancement factor $\mathscr{K}$ (refs 31,32), defined as the maximum optical chirality on the metamaterial surface normalized to the far-field chirality $(\mathscr{K}=\text{Im}[\mathbf{E}\cdot\mathbf{H}^*]_{\text{Nearfield}}/\text{Im}[\mathbf{E}\cdot\mathbf{H}^*]_{\text{Farfield}})$, where the optical chirality follows the same definition as the parameter $C$ defined in ref. 33,34. We show in the following that this second enhancement factor is associated with enhanced local optical density of states that are sensitive to the distinct handedness of molecules, which can be largely tuned and boosted by twisted metamaterials, and suitably exploited for the purpose of molecular chirality detection.

When molecules interact with the near-field of the metamaterial surface, we can model their interaction with the impinging wave by scaling their imaginary part of permittivity and chirality, according to the local field and chiral enhancement factors. This scaling can be understood by considering the enhancement in the power loss density, which stems from the increased local density of states[35]. Since the loss density is directly proportional to $\mathscr{F}\,\text{Im}[\varepsilon]|\mathbf{E}|^2$ and to $\mathscr{K}\,\text{Im}[\kappa_m]\text{Im}[\mathbf{E}\cdot\mathbf{H}^*]$, we can embed the enhancement of local density of states in the effective permittivity $(\mathscr{F}\,\text{Im}[\varepsilon])$ and chirality $(\mathscr{K}\,\text{Im}[\kappa_m])$ coefficients of the molecules. This enhancement factor stems from the near-field interactions between adsorbed molecules and the metamaterial. Recent experiments have demonstrated that the optical chirality coefficient is directly associated with the difference in their absorption of enantiomers (the dissymmetry factor of enantiomers)[33,36]. The twisted metamaterial, with boosted and controllable $\mathscr{K}$, can therefore become an ideal substrate for chiral detection.

**Enhanced CD measurements of enantio-pure molecules**. To prove the unique sensing features of the proposed metamaterial platform, we fabricated a pair of these metamaterials with $+$ and $-60°$ twist angles, as illustrated in Fig. 2a,d, with scanning electron microscope images accordingly shown. Optical

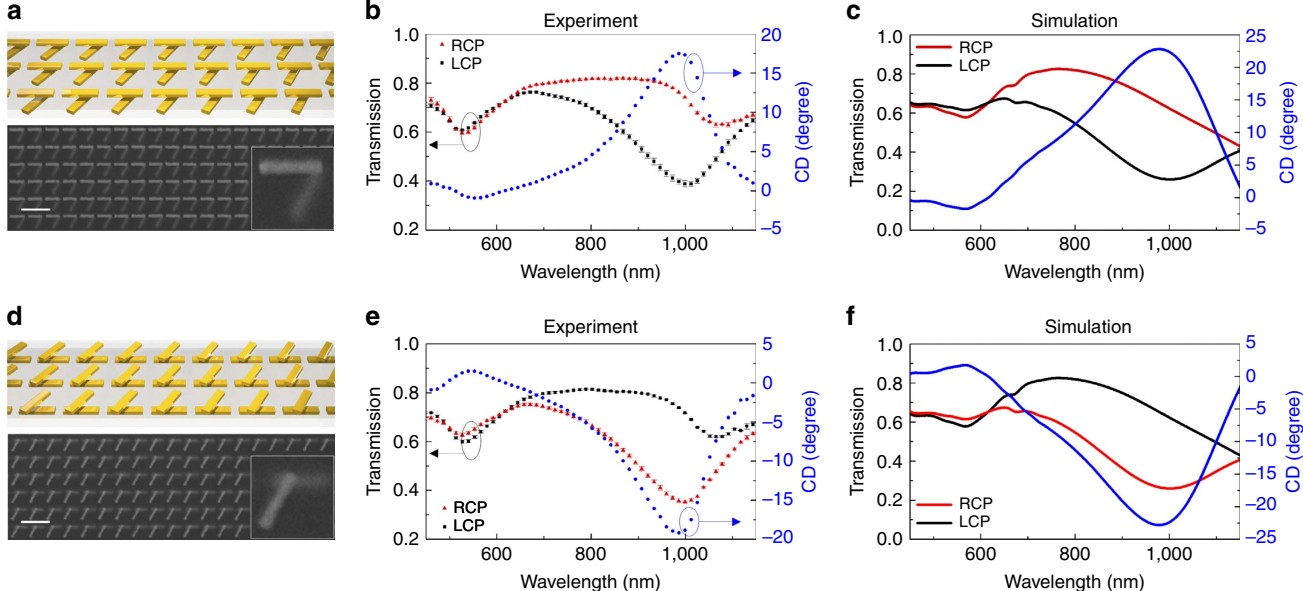

**Figure 2 | Measurements and simulations of an enantiomeric pair of metamaterials. (a)** Scanning electron microscope image and illustration of the $+60°$ metamaterial. **(b)** Experimentally measured transmission (left $y$ axis) of the twisted metamaterial with RCP (red curve) and LCP (black curve) excitation, and its extracted circular dichroism (CD) in degrees (blue curve, right $y$ axis). The error bars correspond to the standard deviation of three measurements at different locations of the metamaterials. **(c)** Corresponding numerical simulations of the background transmission and circular dichroism for the $+60°$ metamaterial. **(d–f)** Figures for the $-60°$ metamaterial. All scale bars are 500 nm. Within the unit cell, each nanorod is 220 nm $\times$ 50 nm $\times$ 40 nm; the square lattice has a dimension of 300 nm $\times$ 300 nm.

measurements of these metamaterials reveal opposite CD, as seen in Fig. 2b,e, and confirmed numerically in panels c and f. In the following we show how such 'enantiomeric pair' of metamaterials can be used to detect molecular chirality.

We first consider a set of well-studied enantiomers, (S)-(+)-1,2-Propanediol and (R)-(−)-1,2-Propanediol. We choose these molecules as a proof-of-principle study, because of easy access to enantio-pure solutions of both enantiomers. We fabricated a pair of +/−60° twisted metamaterials on the same glass substrate, and we created a flow-cell (Fig. 3a) over these metamaterials. The S enantiomers were first filled into the flow-cell, and we collected their corresponding CD spectra on +60° metamaterials and −60° twisted metamaterials, respectively, as shown in Fig. 3b (blue solid circles on +60° twisted metamaterials, blue void circles on −60° twisted metamaterials). The same metamaterial platform was then used to collect the spectra associated with the R enantiomers after thorough cleaning (Fig. 3c, red solid circles on +60° twisted metamaterials, red void circles on −60° twisted metamaterials). The black curves in both panels are control measurements of CD spectra with racemic mixtures. We note that the local interaction between the metamaterial and the molecules induces a small frequency shift near the observed CD maximum, which can be analytically derived from (1) by expanding $CD_o$ around the frequency for which $\partial CD_i / \partial \omega = 0$. As shown in the Supplementary Equation (4), this frequency shift can be potentially used to derive the handedness of the chiral molecules: Supplementary Equation (4) indicates an opposite sign of the frequency shift $\Delta \omega$, associated with opposite signs in $Im[\kappa_m]$ of the enantiomers. However, the magnitude of this shift may be different for left- and right-handed excitations, due to the different chirality enhancement factors from the metamaterial. More importantly, the radiative loss of the metamaterial is relatively broadband, resulting in its large background CD signals that may mask any chiral signals from the analytes, making the frequency shift undetectable.

To remove the large background CD signals from the metamaterial and cancel its effective contribution to $CD_o$, we sum the measurements obtained with the enantiomeric pair of metamaterial substrates holding the same analytes, by taking advantage of the inherent symmetry of the metamaterial pair $CD_i(+60°) = -CD_i(-60°)$. Interestingly, after a suitable calibration process (see Methods: post-processing), we are able to isolate the CD signals associated with the molecules even in the case of small fabrication imperfections that are expected to cause slightly different CD spectra associated with the two metamaterial substrates. The calibration, performed beforehand on the bare metamaterial samples, removes most of the errors stemming from fabrication differences between the two mirror-symmetric metamaterials. What is left after this post-processing is the isolated CD response that directly relates to the sign of the molecule chirality, with an enhanced magnitude that is associated with the near-field interaction between the metamaterial and the chiral molecules (shown in Fig. 3d). In principle, the presence of the chiral molecules may also weakly enhance the CD response of the metamaterial itself. However, this effect is very minor due to the low quantity of molecules involved, and it is not expected to influence the proposed sensing procedure.

Analytically, the enhanced CD response from the molecules can be written as

$$\sum CD_o = \left\{ \mathscr{K}_R^+ \left( |T_{RL}^+|^2 + |T_{LL}^+|^2 \right) \left( |T_{LL}^+|^2 - |T_{RR}^+|^2 \right) \right. $$
$$\left. + \mathscr{K}_L^+ \left( |T_{RL}^+|^2 - |T_{LL}^+|^2 \right) \left( |T_{LR}^+|^2 + |T_{RR}^+|^2 \right) \right\} \frac{4kw}{P^+} Im[\kappa_m], \tag{2}$$

where $P^+ = (|T_{LR}^+|^2 + |T_{RR}^+|^2)^2 + (|T_{LL}^+|^2 + |T_{RL}^+|^2)^2$, the plus apex

indicates the coefficient associated with the +60° metamaterial, and the analogous one for −60° metamaterial may be derived from symmetry considerations: $\mathscr{K}_R^+ = \mathscr{K}_L^-$, $\mathscr{K}_L^+ = \mathscr{K}_R^-$, $T_{RL}^+ = T_{LR}^-$, $T_{LL}^+ = T_{RR}^-$, $T_{LR}^+ = T_{RL}^-$, $T_{RR}^+ = T_{LL}^-$. After post-processing, the near-field enhancement effects associated with the achiral enhancement factor $\mathscr{F}$ are cancelled out, confirming that the near-field chirality enhancement factor $\mathscr{K}$ is the coefficient that has the essential role in detecting enantiomers in our proposed scheme. As discussed in the Supplementary Note 2, this cancellation is robust and, even despite the expected fabrication imperfections, for all measurements reported in the paper we have been able to consistently reproduce the molecular chirality measurements several times using different samples, evidenced by the reported standard deviation in the figures.

To model how our twisted metamaterial geometry regulates this chirality enhancement factor, we use a pair of electric and magnetic dipoles placed within the near field of the metamaterial substrate and monitor their radiated electrical fields with left-handed and right-handed polarizations in the far-field, extracting the output CD. The phases of these two dipoles are properly chosen to resemble the handedness of the enantiomers (see Methods for details). Based on reciprocity, the left-handed or right-handed circularly polarized radiation features of these dipole sources provide an indication of the absorption features of the enantiomers, when excited by circularly polarized waves. With S (Fig. 3e) and R (Fig. 3f) enantiomers placed at 10 nm above the metamaterial substrate, their simulated output CD spectrum matches reasonably well the experimental results, with a pronounced peak near 1,000 nm wavelength. We believe that the discrepancy at shorter wavelengths comes from additional losses in the adhesion layers and the substrate, which are not taken into account in the simulations. The unambiguous detection of the molecular chirality is well summarized by equation (2), where $\Sigma CD_o$ displays opposite signs for S and R enantiomers, due to their opposite signs in $Im[\kappa_m]$. This is experimentally verified in Fig. 3d, showing the summed CD spectrum for the two enantiomers, confirmed by the simulations in Fig. 3g.

**Enhanced CD measurements of other chiral molecules**. We have discussed how the chirality enhancement factor is a near-field effect, and it is, therefore, especially sensitive to very thin layers of molecules close to the metamaterial surface, which is of most practical interest in applications requiring high sensitivity chirality detection. In order to prove this point, we tested a monolayer protein sample Concanavalin A prepared at a concentration of 1 mg ml$^{-1}$ in a buffer solution (see Methods: Preparation of chemicals for details). The protein was spin-coated on the metamaterials, forming a 10 nm thick film. This thickness was confirmed through ellipsometry measurement. Figure 4a shows the initial negative bending near 970 nm in the measured $\Sigma CD_o$, indicating the protein's right-handedness[37]. Although a much smaller amount of analytes was used in this experiment compared to the one in Fig. 3, the results in Fig. 4a show a much cleaner spectrum, attributed to the bigger intrinsic molecular chirality of Concanavalin A. Further verification was conducted on a chiral anticancer drug, Irinotecan Hydrochloride, ((S)-4,11-diethyl-3,4,12,14-tetrahydro-4-hydroxy-3,14-dioxo-1H-pyrano[3′,4′:6,7] indolizino[1,2-b]quinolin-9-ylester, Sigma Aldrich), which is commonly used to treat colorectal cancer or metastatic cancers that chemotherapy has failed to treat. The resultant $\Sigma CD_o$ clearly shows an initial positive bending over the spectral range between 750 to 1,050 nm, indicating its left-handed nature (Fig. 4b). For large chiral molecules, a net Cotton effect is commonly observed due to the superposition of individual Cotton effects. In Fig. 4, we indeed observe the Cotton effect, consistent with a conventional

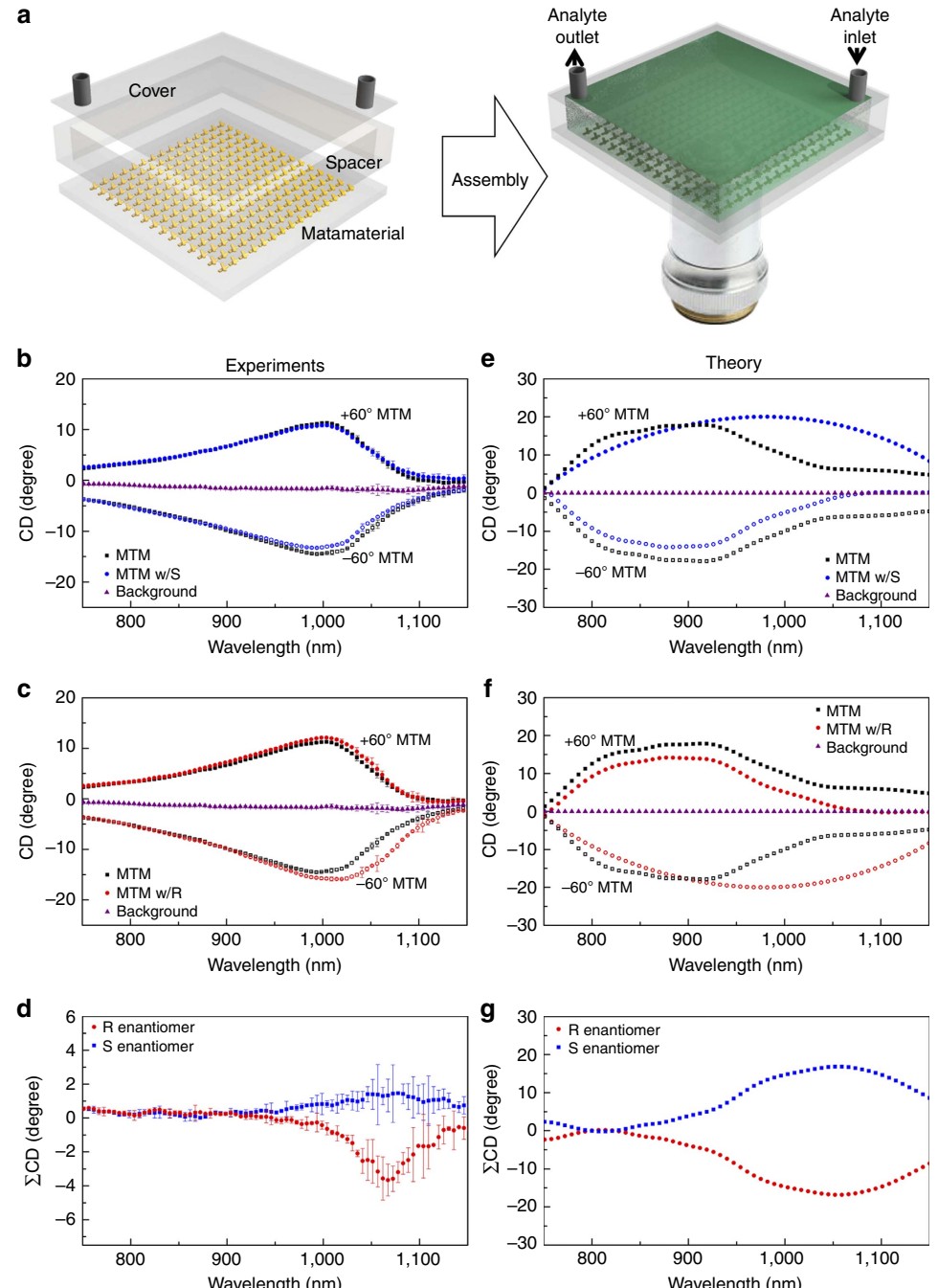

**Figure 3 | CD spectra of enantiomers on metamaterial substrates.** (**a**) The flow-cell experimental set-up. (**b**) Experimental measurement of (S)-(+)-1,2-Propanediol on ±60° metamaterials (MTM). The blue solid circles refer to the S enantiomer on the +60° metamaterial, the blue void circles refer to the S enantiomer on the −60° metamaterial. Similarly, the black curves refer to ±60° metamaterials with the racemic mixture (formed with 1:1 ratio of right- and left-handed Propanediol). The purple triangles indicate the center line of the left-handed and right-handed metamaterials loaded with the racemic mixture (calculated by summing the two black curves and taking its half). The center line does not exactly fall to zero, due to small fabrication imperfections that make the ±60° metamaterials not completely symmetric. The error bars indicate the standard deviation, which comes from three repeated measurements for each analyte across different locations on the metamaterials. 'MTM w/S' denotes metamaterials with S enantiomers. (**c**) Experimental measurement of (R)-(−)-1,2-Propanediol on +60° (red solid circles) and −60° (red void circles) metamaterials. The black and purple curves follow the same nomenclatures as in **b**. The error bars indicate the standard deviation from three repeated measurements for each analyte across different locations on the metamaterials. 'MTM w/R' denotes metamaterials with R enantiomers. (**d**) CD summation to remove the background CD of the metamaterials. The curves show clear opposite signs for R and S enantiomers. The error bars indicate the standard deviation from the three repeated measurements of each analyte from **b**,**c**. (**e**) Full wave numerical simulations based on our model for conditions in **b**. (**f**) Full wave numerical simulations for conditions in **c**. (**g**) Calculation of CD summation for the S and R enantiomers on the metamaterials.

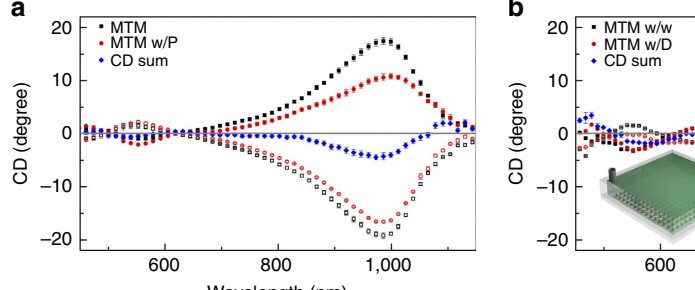

**Figure 4 | Monolayer chiral sample measurements.** (**a**) Measured monolayer protein (Concanavalin A prepared with 1 mg ml$^{-1}$ concentration) on ±60° metamaterials (MTM). The solid black (red) squares (circles) refer to measurements from the +60° metamaterials (with proteins), the void black (red) squares (circles) refer to that from the −60° metamaterials (with proteins), and the grey straight line indicates 0 degree CD for guide-to-the-eye. The CD summation of the protein (blue diamonds) shows a negative bend, indicating the overall 'right-handedness' of the analyte. For a large protein with different chiral centers, the CD signal is more complicated, and may show spectral oscillations with changing signs. Our results are consistent with chirality assignment based on conventional CD measurements. The error bars indicates standard deviation from three repeated measurements for each analyte across different locations on the metamaterials. The legend 'MTM w/P' denotes the metamaterials with proteins. (**b**) CD measurements of a chiral anticancer drug (Irinotecan hydrochloride at a concentration of 1 mg ml$^{-1}$) on metamaterials. Its CD summation (blue diamonds) with the same pair of metamaterials shows positive bend, indicating its overall 'left-handedness'. The black curves were measurements from the ±60° metamaterials with water, and the grey straight line indicates 0 degree CD for guide-to-the-eye. The error bars indicates standard deviation from three repeated measurements for each analyte across different locations on the metamaterials. The legend 'MTM w/w' denotes the metamaterials with water; 'MTM w/D' denotes the metamaterials with the chiral drug. The inset indicates that this measurement is conducted in a flow-cell setting.

CD measurement of the molecule (see Supplementary Note 4). In our enhanced CD spectrum, this effect is transferred to the visible spectrum through plasmon-induced chirality transfer[12,15]. The implications of these additional examples are twofold. First, the proposed method shows an inherent robustness to molecules with different molecular weight, solution quantity, buffer solution, and it is only sensitive to the handedness of the chiral molecules. Second, the enhanced chiral detection is observed with a monolayer of analytes, confirming that our scheme relies purely on a near field effect (Supplementary Note 5).

## Discussion

In a broader context, our results show that, by employing a pair of twisted metamaterials we can realize a powerful platform for chiral molecular detection. We experimentally proved drastically enhanced sensitivity to molecular chirality, sufficient to detect as low as ∼55 zeptomoles of molecules in our imaging area, which corresponds to ∼44 molecules per unit cell of the metamaterial (see Methods: analyte quantity estimation). Please note that such sensitivities are not achieved solely due to the large magnification objectives, therefore, they should not be considered the best possible performance for the described sensing procedure. With proper instrument development (detectors with low noise or smaller pixels), a higher sensitivity is expected. Increased sensitivity can also be achieved by incorporating a nanofluidic system and flowing chiral molecules between the twisted layers instead of atop. We envision that our platform may be directly integrated with such nanofluidic systems[38], allowing one to detect attograms of analytes per measurement, and to separate and detect in real-time the level of chirality down to a few molecules. The molecules that we have chosen for these proof-of-principle studies cover a variety range of molecular weights from 76.09 Da (or 76.09 gmol$^{-1}$) to 104 kDa, indicating the robustness of the method. More importantly, our experiments demonstrate that the specific use of a pair of twisted metamaterials with opposite rotation allows us to suppress the background CD response from the metamaterials and to isolate the molecular chiral response to univocally reveal its handedness. Although our method still measures cumulative optical activity, sharing a similar drawback as conventional CD measurements that cannot provide specific

information on chiral centers, our measurements take only a fraction of a second, orders of magnitude faster than currently available commercial equipment for CD measurements. Our approach is ideal for sequential measurements using microfluidic systems of very small quantities of molecules. We believe that these findings can open important directions for biomedical measurements and drug synthesis, based on the unique properties of suitably engineered optical metamaterials.

## Methods

**Sample fabrication.** The metamaterial sample was fabricated using electron beam lithography and an etch-back planarization method initially introduced in ref. 19 on an optical flat glass substrate. The spacer between the two metasurfaces was achieved using electron beam deposition of silicon dioxide, and the thickness was controlled as 80 nm with a 5% thickness variation. The dimensions of the unit cell nanodipole is 220 nm × 50 nm × 40 nm (length × width × thicknesss). The unit cell is embedded in a square lattice of 300 nm × 300 nm. The fabricated plasmonic metamaterial sample has a foot-print of 200 μm by 200 μm, which includes more than 400,000 unit cells.

**Preparation of chemicals.** Enantiomers (S)-( + )-1,2-Propanediol and (R)-( − )-1,2-Propanediol were used as received (Sigma-Aldrich, products 540242 and 540250, 96%). The Propanediol enantiomers were prepared with a flow-cell with thickness of ∼70 μm. The anticancer drug (Sigma-Aldrich, Irinotecan hydrochloride) was dissolved in water and formed a solution with concentration of 1 mg ml$^{-1}$. The anticancer drug experiment was also performed with the flow-cell. The flow-cell was prepared and replaced every time for each measurement. The protein was dissolved in 10 mM Tris/HCl buffer solution with a controlled pH value at 7, forming a concentration of 1 mg ml$^{-1}$. The prepared protein solution was spin-coated onto the clean metamaterial sample at a spin speed of 2,000 r.p.m., forming a monolayer of ∼10 nm, which is confirmed with the ellipsometry measurements (J.J. Woollam M-2000 DI).

**Cleaning procedure.** The metamaterial samples were used for multiple measurements. After each measurement, the sample was immersed in deionized water for 72 h and then cleaned in base piranha solution to remove excess organic residues on the sample (3:1 mixture of ammonium hydroxide (NH$_4$OH) with hydrogen peroxide). This treatment also left the metamaterial hydrophilic for better adhesion especially for the protein sample, which was prepared by spin-coating.

**Optical measurement.** The transmission spectrum was measured using a home-built set-up. A tungsten halogen lamp (Ocean Optics, HL-2000-HP) was used as a white light source (360–2,400 nm) at a power of 8.8 mW. The output from the halogen lamp was coupled to a spectrometer with an optical fibre, directly followed by a 35 mm focal length lens (AC254-035-B), a linear polarizer (Newport, 10LP-

VIS-B) and a quarter wave-plate (Special Optics, 8-9012-1/4) to obtain circularly polarized incident light. This incident light was focused onto the sample with a $10 \times$ objective (N.A. = 0.28, Mitutoyo) at normal incident angle. The focal spot on the sample was around 0.5 mm in diameter, which covered the entire sample area (200 μm by 200 μm). The transmitted light was then collected via a second objective ($100 \times$, N.A. = 0.70, Mitutoyo), followed by an imaging lens with a focal length of 200 mm (Thorlabs, AC254–200-B-ML) before the spectrometer (Princeton Instruments, SpectraPro-500i). We then selected a small area of the magnified image using the entrance slit (about 2.6 mm by 0.26 mm) of the spectrometer corresponding to an area of 26 μm by 2.6 μm on the sample. The transmitted signal was dispersed by a grating (300 g mm$^{-1}$, 500 nm blazing) inside the spectrometer onto a liquid nitrogen-cooled, Si charge coupled device (Princeton Instruments, 7508-0002). The displayed transmission spectra are combined spectra taken at three different central wavelengths with an integration time of 100 ms for each measurement. Three to five measurements were repeated on each sample across the metamaterial from top to bottom and the results were found to be consistent.

**Analyte quantity estimation.** To estimate how many ConA molecules cover the imaging area, we used a silicon surface as the reference surface for molecule counting. We spin-coated the wafer with the ConA solution under the same condition as we prepare our sample (2,000 r.p.m. at a concentration of 1 mg ml$^{-1}$). We scanned the spin-coated surface using atomic force microscope. The atomic force microscope images before and after scanning indicate relatively densely packed molecules on the surface (Supplementary Fig. 4 and Supplementary Note 6). Using image-processing software (ImageJ), we estimate that approximately $44 \pm 7$ molecules cover one unit cell area of the metamaterial. The imaging area is 26 μm by 2.6 μm on the sample, which covers $\sim$751 unit cells of the metamaterial, corresponding to approximately 33,360 molecules in the imaging area, or $\sim$55 zeptomoles. The area that one ConA molecule occupies is estimated based on the dimension of the molecule. ConA is a homotetramer with a dimension of $6.3 \times 8.7 \times 8.9$ nm determined by X-ray crystallography, which indicates the most compact structural dimension[39].

**Full-wave numerical simulations.** Fully vectorial numerical simulations are conducted using commercially available software based on finite integration method (CST Microwave Studio 2011). The material permittivities used in the simulations are explicitly described as following: permittivity of gold follows the values from ref. 40; the permittivity of the adhesive layer, Titanium, is simulated using values obtained from website (http://refractiveindex.info/); the glass substrate is considered as a nondispersive silicon dioxide with permittivity of 2.25 and negligible absorption. In the numerical calculations of Fig. 3, the chiral molecules are modelled using a pair of electric (with amplitude of $3.54 \times 10^{-31}$ cm) and magnetic dipoles (with amplitude of $1.06 \times 10^{-22}$ Am$^2$) with in and out of phase for right-handed (R) and left-handed (S) enantiomers, respectively. The racemic mixture is simulated when the electric and magnetic dipoles are perpendicular to each other. The dipole pairs are placed at 10 nm above the twisted metamaterial substrate in the center of the crossed rods. CD spectra are calculated by extracting the ratio between the left-handed and right-handed radiated fields in the far field.

**Post processing.** With fabrication imperfections in reality, $CD_i^+$ from the $+60°$ metamaterial doesnot completely cancel out $CD_i^-$ from the $-60°$ metamaterial (as shown in Fig. 3). But in the summation of CD,

$$\sum CD_o = \frac{(CD_o^+ - CD_i^+) + (CD_o^- - CD_i^-)}{4kw\,\mathrm{Im}[\kappa_m]}P^+$$
$$= \mathscr{K}_R^+ \left(|T_{RL}^+|^2 + |T_{LL}^+|^2\right)\left(|T_{LR}^+|^2 - |T_{RR}^+|^2\right)$$
$$+ \mathscr{K}_L^+ \left(|T_{RL}^+|^2 - |T_{LL}^+|^2\right)\left(|T_{LR}^+|^2 + |T_{RR}^+|^2\right),$$

it can remove the system errors that come from this fabrication imperfection because the same imperfection exists in $CD_o^+$ and $CD_o^-$ in all measurements with and without molecules. Therefore, $\Sigma CD_o$ is still directly related to the signs of the molecular chirality. To clearly demonstrate this and de-embed a CD sum that reflects the molecular property free from the fabrication imperfections, we subtract the CD sum of each enantiomer by the background CD sum as shown in the above equation.

**Data availability.** The data that support the findings of this study are available from the corresponding author upon request.

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

## Acknowledgements

This work was supported by the Welch Foundation with grant numbers F-1802 (A.A.), F-1662 (X.L.), the Air Force Office of Scientific Research with grant number FA9550-13-1-0204 (A.A.), NSF DMR-1306878 (X.L.), and the Army Research Office W911NF-11-1-0447 (A.A., X.L.).

## Author contributions

A.A. and Y.Z. conceived the idea and designed the experiments. Y.Z. and A.N.A. performed the theoretical and numerical calculations. Y.Z. and L.S. performed the experiments, with assistance from J.S.. X.L. supervised the experiments. A.A. supervised the theoretical and numerical calculations. All authors contributed to the analysis of the data and writing of the manuscript.

## Additional information

**Competing financial interests:** The authors declare no competing financial interests.

**Publisher's note**: 

