## [Peer Review File · Nature Communications]

Reviewers' comments:

Reviewer #1 (Remarks to the Author):

The manuscript describes the use of chiral plasmonic nanostructures to achieve ultrasensitive chiral (stereochemical) detection. This phenomenon was first reported by Hendry et al (2010) *Nature Nanotechnology*, and as subsequently been reported by Karimullah et al 27 (2015) 5610 *Adv Materials*, Tulus et al 137 (2015) 83806 *J Am Chem Soc* and by Khoo et al 6 (2016) 19658 *Scientific Reports*.

Unfortunately given the previous work in the field I cannot recommend the manuscript for publication in *Nature Comm*, on the grounds of lack of novelty, as the work reported is an incremental advancement on these previous studies. The work neither provides new insight into the phenomenon of interaction of chiral evanescent fields with chiral (bio)materials, nor does it demonstrate a new application or capability of chiral fields. Either of these two would warrant publication in *Nat Comm*

The authors do claim zeptomole sensitivity, which has not been previously reported, however this sensitivity arises from the use of a higher magnification objective to that used in previous work (i.e. due to an instrument develop rather than new science).

I have no major issues with the experimental data or the conclusion the authors draw from them. However, I think the work would be more appropriate in a Journal such as *ACS Photonics*.

Before the work is published anywhere I suggest the authors include in the manuscript itself modelling showing the optical chirality of the near fields around the structures.

Reviewer #2 (Remarks to the Author):

The authors investigated a possibility of improving the sensitivity in the detection of a molecule's chirality by using a twisted metamaterial platform. In fact, there have been quite a few works that tried to measure a molecule's CD by using a chiral metamaterial. However, this work newly proposes a very simple yet powerful method of measuring a molecule's CD with two chiral metamaterials having opposite CDs. By analyzing obtained data, one can extract the molecule's CD without being interfered by large CD of a chiral metamaterial.

The paper is well organized and well written. I personally recommend the publication of this work in *Nature Communications*. Minor questions are as follows;

(1) If two chiral metamaterials are fabricated erroneously to possess slightly different absolute values of chirality, what will be effect of this in the extraction of a molecule's CD?

(2) What is the relationship between chirality enhancement and the distance between the metamaterial and the molecule?

Reviewer #3 (Remarks to the Author):

The manuscript "Chirality Detection of Enantiomers Using Twisted Optical Metamaterials", by Yang Zhao et al. reports on enhanced sensing of chiral molecules using "twisted" metamaterials. The main concern is that similar reports have been made before. In fact, Kadodwala's group made similar claims, explained with the same physical processes - optical chirality. Novelty is therefore an issue.

In the abstract: "enantiomers of different chirality" is redundant.

The statement "single enantiomer drugs are more effective than their racemic mixtures" is wrong. Overall, the introduction is overdramatic and misleading. The vast majority of enantiomeric pairs in pharmaceuticals differ in potency only. Toxicity is an issue in a few cases. The fact that enhanced CD detection would help in the synthesis of new chemical compounds is true and it is very well presented in the introduction.

The paper is not well written. Several passages are unclear, due to sentences that are too long and due to confusing grammar. These are examples of sentences that need to be clarified:

- "scaling their imaginary part of permittivity and chirality according to the field and chiral enhancement factors sampled at the molecule location"
- "enhancement in power loss density stemming from the increased local density of states"
- "we can embed the local density of states enhancement in effective imaginary permittivity and chirality coefficients of the molecule layer that include the enhancement factor stemming from their near-field interactions."
- "However, the magnitude of this shift may be different for left-handed and right-handed excitations due to the different chirality enhancement factors from the metamaterial; more importantly, the metamaterial causes a large loss factor in o CD, resulting in its large background CD signals that risks to overwhelm in practice any chiral signal from the analytes, making the frequency shift easily overlooked."

The science is not well explained. There is no physical explanation for the claim that the effective chirality can be boosted by two distinct mechanism. Instead, there is too much explanation about simple things, such as adding two opposite spectra to get zero. It is not clear why optical chirality would not also enhance the CD of the meta-molecules. For clarity, the authors should use well established notations for the chirality parameter, optical chirality and dissymmetry factor. The physical processes of their interactions should be clearly stated.

The figures are mostly understandable and are of good quality. Some minor remarks:

- Figure 1 is fine
- Figure 2, the dimensions of the nanostructures should be specified. Having a 500 nm scale bar is not helpful.
- Figure 3, it took me a long time to understand that the left side is experiment and the right is theory. This should be clarified on the figures themselves. The CD spectrum of the molecules should be provided.
- Figure 4, the CD spectrum for the molecules should be provided.

The data do not support the claims very well.

- The difference in the experimental spectra in figure 3 is not very large. How reproducible is this? The authors say that it is in the text but they do not show any evidence in the manuscript or in the supporting information.
- The authors should measure more molecules and give evidence that their method works. Kadodwala's group investigated a whole range of molecules.
- This work claims the use of a flow cell, but it is not clear whether the molecules were flown through the cell while measured or whether the authors just filled the cell up and then measured.
- In figure 4, the authors claim that the CD changes because this is typical of large molecules with several chiral centres. This seems rather naïve. In principle, the CD response can be accurately calculated for simple molecules such as amino acids. When the molecules have more than one chiral centre it becomes much more difficult. For example, the CD spectrum of proteins was never completely understood. Yes, CD changes according to resonances near chiral centres but often the resonances overlap for all centres. A general rule is that (at least for simple molecules) the CD effect changes sign (bisignate) if it is due to exciton coupling (through space coupling between 2 chromophores). Here, the authors do not show the CD of the molecules whereas, in Figure 2, the

CD of the nanostructures does show a sign change depending of wavelength.

Response to the Reviewers' Comments on the Manuscript NCOMMS-16-01155A-Z entitled "Chirality Detection of Enantiomers Using Twisted Optical Metamaterials" submitted to Nature Communications

We would like to thank the reviewers for carefully reading our paper. We are very encouraged by Reviewer 2's recognition of our scientific contribution and of the impact of our work. We also appreciate the comments and criticisms from Reviewers 1 and 3. The valuable comments from all reviewers have allowed us to significantly improve our manuscript. In the following, we include a point-by-point response to the questions and comments of each reviewer. With this resubmission, we also attach a revised manuscript that includes the updated text to address the reviewers' concerns. We do hope that the reviewers and editor may appreciate our significant effort in addressing these questions, and find our revised manuscript an exciting contribution to the community that may deserve publication in Nature Communications.

Reviewer #1:

The manuscript describes the use of chiral plasmonic nanostructures to achieve ultrasensitive chiral (stereochemical) detection. This phenomenon was first reported by Hendry *et al* (2010) Nature Nanotechnology, and as subsequently been reported by Karimullah *et al* 27 (2015) 5610 Adv Materials, Tulus *et al* 137 (2015) 83806 J Am Chem Soc and by Khoo *et al* 6 (2016) 19658 Scientific Reports.

Unfortunately given the previous work in the field I cannot recommend the manuscript for publication in Nature Comm, on the grounds of lack of novelty, as the work reported is an incremental advancement on these previous studies. The work neither provides new insight into the phenomenon of interaction of chiral evanescent fields with chiral (bio)materials, nor does it demonstrate a new application or capability of chiral fields. Either of these two would warrant publication in Nat Comm.

The authors do claim zeptomole sensitivity, which has not been previously reported, however this sensitivity arises from the use of a higher magnification objective to that used in previous work (i.e due to an instrument develop rather than new science).

I have no major issues with the experimental data or the conclusion the authors draw from them. However, I think the work would be more appropriate in a Journal such as ACS Photonics.

Before the work is published anywhere I suggest the authors include in the manuscript itself modelling showing the optical chirality of the near fields around the structures.

Authors: We thank the reviewers for mentioning a few pioneering works in the use of chiral metasurfaces to sense molecular chirality. Here we compare our work with these papers one by one and point out the novelty of our work.

The first chiral detection using a metasurface platform was published by Hendry *et al* (2010) Nature Nanotechnology. In this paper, a gammadion structure is used, and they studied 6 proteins, including myoglobin, haemoglobin, bovine serum albumin, β -lactoglobulin, outer membrane protein A (Omp A), and concanavalin A. The main mechanism of this work relies on a

spectral shift in the far field due to the nearfield interactions between the chiral molecules and the metasurface.

The subsequent publications that are mentioned by the reviewer, (Karimullah *et al* Adv Materials (2015), Tullius *et al* J Am Chem Soc (2015), Khoo *et al* Scientific Reports (2016)), are all based on the same mechanism that was first reported by Hendry *et al*. Specifically, Karimullah *et al* used a “shuriken” metasurface to study the chiral detection of Concanavalin A; Tullius *et al* used a “shuriken” metasurface to study two proteins: 5-enolpyruvylshikimate 3-phosphate synthase (EPSPS) and Shikimate kinase (SK). Khoo *et al* used a “gammadion” metasurface to study actin molecules and filaments. These publications are related to our work only in a big picture sense: they use plasmonic metamaterials to detect the handedness of chiral molecules.

In these existing publications, the metasurfaces are a single layer gammadion (or “shuriken”) structure with C4 symmetry. Near field interactions between the molecules and the metasurface propagate to the far-field, and are detected as a spectral shift in the collective circular dichroism spectrum of the metasurface-molecule assembly.

However, this mechanism has intrinsic limitations in achieving high sensitivities, as we point out in our paper. The limitations can be fundamentally explained from Poynting’s theorem and reciprocity theorem, as we detailed in our manuscript in page 5-6. By analyzing the loss density, two enhancement mechanisms contribute together to all these existing works. The first one is near field enhancement, the other is chiral enhancement. Near field enhancement has been widely used for refractive index sensing, it is known that new molecules within the nearfield of the plasmonic metasurfaces results in a spectral shift regardless of their handedness (e.g. Kabashin *et al*, *Nature Materials* 8, 867 (2009); Im *et al*. *Nature Biotechnology* 32, 490 (2014); Cetin *et al*, *Light: Science & Applications* 3, e122, (2014); etc.). However, such an achiral-induced-spectral-shift can off-set the detected spectrum and introduce an extra uncertainty to the chiral detection. To achieve higher sensitive detection, such effect has to be removed from the final spectra.

Very different from these publications, our work is innovative in three aspects which overcome the abovementioned limitations:

1) Our mechanism **doesn’t rely on any spectral shift** to detect the handedness the molecules, but an absolute sign difference in the $\sum CD$ signal. In our approach, we use a pair of chiral metamaterials for the purpose of eliminating the unwanted nearfield enhancement. Specially, we measure the same chiral molecules separately with the pair of metamaterials, then sum their spectra ($\sum CD$). Since the chiral molecules introduce the same refractive index to both metamaterials (please refer to detailed derivations in Supplemental Information 1.2), we can cancel the unwanted near field enhancement (\mathcal{E}) by summing the CD spectra, which eliminate any possible contribution from refractive index change that may confuse the spectral shift. In addition, this simple approach also removes the broadband background from the metamaterials, but still takes the advantage of the chiral enhancement, purely reveals the handedness of the molecules.

2) Our multilayer chiral structure can amplify the chiral-chiral interaction between the metamaterial and the molecules to the far field (10 deg vs 0.1 deg). In addition, our design provides extra flexibility to easily tune the enhanced circular dichroism to a desired frequency, where the solvent may exhibit minimal absorption. The tunability can be achieved by adjusting the relative angles between the layers of the structure (Figure S1 in Supplemental Information),

rather than changing the thickness of structures. Changing the thickness could eventually affect the performance of the metasurface due to Ohmic losses in metallic nanostructures.

3) As the reviewer mentioned, Zeptomole chiral detection has not been reported, especially with label-free methods. In this work, we achieved Zeptomole sensitivity in a label-free method. And mostly importantly, we demonstrate that our approach works for a large range of molecular weight, from 76.09Da (1,2-propanediol) to 100kDa (ConA).

Please note that the measured levels are not at the fundamental limit of our method. With proper instrument development (detectors with low noise or smaller pixels), a much higher sensitivity is expected with our mechanism. With this summary, we hope that we have convinced the reviewer that we have “demonstrated a new capability of chiral fields” in this work.

Finally, we would like to thank the reviewer for suggesting the relevance of optical chirality of the near fields around the structure, in the revised manuscript we added this information in our supplemental information with an additional section (1.5, Figure S3a). In addition, we have included the references suggested by this reviewer that were not already cited in the original manuscript.

Reviewer #2:

The authors investigated a possibility of improving the sensitivity in the detection of a molecule's chirality by using a twisted metamaterial platform. In fact, there have been quite a few works that tried to measure a molecule's CD by using a chiral metamaterial. However, this work newly proposes a very simple yet powerful method of measuring a molecule's CD with two chiral metamaterials having opposite CDs. By analyzing obtained data, one can extract the molecule's CD without being interfered by large CD of a chiral metamaterial.

The paper is well organized and well written. I personally recommend the publication of this work in Nature Communications.

Authors: We thank the reviewer for his/her careful review and for highlighting some novel aspects of our work. We are greatly encouraged by his/her positive comments and the acceptance decision.

Minor questions are as follows;

(1) If two chiral metamaterials are fabricated erroneously to possess slightly different absolute values of chirality, what will be effect of this in the extraction of a molecule's CD?

Authors: We thank the reviewer for raising this very good question. Most of the difference in fabrication can be corrected through our calibration and post-processing, which we called a “de-embedding process” in the paper. However, to completely remove this slight difference is nontrivial, because it also affects the strength of the chiral enhancement. As a result, the magnitude of the $\sum CD$ signals for S and R enantiomers will be slightly different. As an example in our case, the bare -60° metamaterial has a relatively stronger signal as shown in Figure 2. This indicates that $\mathcal{K}_S^- > \mathcal{K}_R^+$, resulting in a final $\sum CD$ of the R-enantiomers (or all right-handed molecules) with a bigger magnitude compared to their left-handed enantiomeric pair at the same concentration.

This error can be best calibrated with a known molecule at a known quantity, such as using a nanofluidic channel integrated with the metamaterial sensor to precisely control the amount of chemicals. However, please note that, even with this small difference, $\sum CD_0$ is still directly related to the *signs* of the molecular chirality. We used Figure 3 as our calibration of the handedness of the molecules, this is how we predict the handedness of the rest of the molecules.

What is the relationship between chirality enhancement and the distance between the metamaterial and the molecule?

Authors: The enhancement decays exponentially away from the surface. In the revised manuscript, we have added an extra figure in the supplemental information where the enhancement factor decays exponentially away from the surface (Fig. S3 b, c). This question is related to the near field chirality distribution question raised by Reviewer 1. We have added both information in Supplemental Information section 1.5 and Figure S3 accordingly.

Reviewer #3:

The manuscript "Chirality Detection of Enantiomers Using Twisted Optical Metamaterials", by Yang Zhao et al. reports on enhanced sensing of chiral molecules using "twisted" metamaterials. The main concern is that similar reports have been made before. In fact, Kadodwala's group made similar claims, explained with the same physical processes - optical chirality. Novelty is therefore an issue.

Authors: We respectfully disagree with the reviewer's comment on the novelty of our work. In the following we cite relevant papers from Kadodwala's group and compare them one by one to point out the novelty of our work.

The first chiral detection using a metasurface platform was published by Hendry et al (2010) Nature Nanotechnology. In this paper, a gammadion structure is used, and they studied 6 proteins, including myoglobin, haemoglobin, bovine serum albumin, β -lactoglobulin, outer membrane protein A (Omp A), and concanavalin A. The main mechanism of this work relies on a **spectral shift in the far field** due to the nearfield optical chirality of the metasurface.

The subsequent publications that are mentioned by the reviewers, (Karimullah et al Adv Materials (2015), Tullius et al J Am Chem Soc (2015), Khoo et al Scientific Reports (2016)), are all based on the same mechanism that was first reported by Hendry et al. Specifically, Karimullah et al used a "shuriken" metasurface to study the chiral detection of Concanavalin A; Tullius et al used a "shuriken" metasurface to study two proteins: 5-enolpyruvylshikimate 3-phosphate synthase (EPSPS) and Shikimate kinase (SK). Khoo et al used a "gammadion" metasurface to study actin molecules and filaments. These publications are related to our work in the big picture: using plasmonic metamaterials to detect the handedness of chiral molecules.

In these existing publications, the metasurfaces are a single layer gammadion (or "shuriken") structure with C4 symmetry. Near field interactions between the molecules and the metasurface propagate to the farfield, are detected as a spectral shift in the collective circular dichroism spectrum of the metasurface-molecule assembly.

However, this mechanism has intrinsic limitations in achieving high sensitivities.

The limitations can be fundamentally explained from Poynting's theorem and reciprocity theorem, as we detailed in our manuscript. By analyzing the loss density, two enhancement mechanisms contribute together to all these existing works. The first one is near field enhancement, the second one is chiral enhancement. However, it is well known that near field enhancement has been widely used for refractive index sensing, which results in a *spectral shift* when new molecules are introduced to the plasmonic metasurfaces regardless of their handedness. Such spectral shift introduces extra noises to the spectra, thus limits the detection sensitivity, therefore has to be removed from the final spectra if higher sensitivity is desired. A second issue that limits the detection sensitivity for the existing works comes from the structural design of the metasurfaces. Because all the detections are based on farfield circular dichroism spectrum, the intrinsic low quality factor of the radiative modes from these metasurfaces, in addition to the weak farfield chiral effect for such chiral metasurfaces together, pose extra constraints on high sensitivity detection.

Very different from these publications, our work is innovative in three aspects which overcome the abovementioned limitations:

1) Our mechanism **doesn't rely on any spectral shift** to detect the handedness of the molecules, but an absolute sign difference in the sumCD signal. This simple approach eliminates the broadband background from the metamaterials, but still takes the advantage of the chiral enhancement, purely reveals the handedness of the molecules. In particular, by summing the CD spectra, we also eliminate any possible contribution from refractive index change that may confuse the spectral shift.

2) Our multilayer chiral structure intrinsically provides a strong chiral effect, which can amplify this chiral-chiral interaction between the metamaterial and the molecules to the far field. In addition, our design provides the flexibility to easily tune the enhanced circular dichroism to a desired frequency, where the solvent may exhibit minimal absorption. Our tunability relies on a shape change (Figure S1 in Supplemental Information), rather than a change of thickness, therefore provides a possibility to integrate multipole arrays on the same platform.

3) As Reviewer 1 mentioned, Zeptomole chiral detection hasn't been reported, especially with label-free method. In this work, we achieved Zeptomole sensitivity in a label-free method. And mostly importantly, we demonstrate that our approach works for a large range of molecular weight, from 76.09Da (1,2-propanediol) to 100kDa (ConA).

In the abstract: "enantiomers of different chirality" is redundant.

Authors: per the reviewer's suggestion, we removed the term "of different chirality" after "enantiomers" to avoid redundancy.

The statement "single enantiomer drugs are more effective than their racemic mixtures" is wrong. Overall, the introduction is overdramatic and misleading. The vast majority of enantiomeric pairs in pharmaceuticals differ in potency only. Toxicity is an issue in a few cases.

Authors: We respectfully disagree with the reviewer's comment. Because for the same amount of active ingredient, single enantiomer drugs reduce the total given dose. This statement is supported by many existing publications, such as: Agranat et al, Nature Reviews Drug Discovery

1, 753, 2002; McConathy et al, Prim Care Companion J Clin Psychiatry 5, 70, 2003; just to name a few.

To avoid confusion, we have slightly revised this sentence. It now reads "single enantiomer drugs are *oftentimes* more *efficient* than their racemic mixtures"

The fact that enhanced CD detection would help in the synthesis of new chemical compounds is true and it is very well presented in the introduction.

Authors: We thank the reviewer for this positive comment.

The paper is not well written. Several passages are unclear, due to sentences that are too long and due to confusing grammar. These are examples of sentences that need to be clarified:

Authors: We have made these long sentences shorter or added proper punctuation for clarity.

- "scaling their imaginary part of permittivity and chirality according to the field and chiral enhancement factors sampled at the molecule location"

Authors: This sentence now reads "scaling their imaginary part of permittivity and chirality, according to the local field and chiral enhancement factors".

- "enhancement in power loss density stemming from the increased local density of states"

Authors: This sentence now reads "enhancement in the power loss density, which stems from the increased local density of states".

- "we can embed the local density of states enhancement in effective imaginary permittivity and chirality coefficients of the molecule layer that include the enhancement factor stemming from their near-field interactions."

Authors: This sentence now reads "we can embed the enhancement of local density of states in the effective permittivity ($\mathcal{I} \text{Im}[\epsilon]$) and chirality ($\mathcal{K} \text{Im}[\kappa_m]$) coefficients of the molecules. This enhancement factor stems from the near-field interactions between the molecules and the metamaterials."

- "However, the magnitude of this shift may be different for left-handed and right-handed excitations due to the different chirality enhancement factors from the metamaterial; more importantly, the metamaterial causes a large loss factor in o CD, resulting in its large background CD signals that risks to overwhelm in practice any chiral signal from the analytes, making the frequency shift easily overlooked."

Authors: This sentence now reads "However, the magnitude of this shift may be different for left-handed and right-handed excitations, due to the different chirality enhancement factors from the metamaterial. More importantly, the radiative loss of the metamaterial is relatively broadband, resulting in its large background CD signals that may mask any chiral signals from the analytes, making the frequency shift undetectable."

Reviewer: The science is not well explained. There is no physical explanation for the claim that the effective chirality can be boosted by two distinct mechanism. Instead, there is too much explanation about simple things, such as adding two opposite spectra to get zero.

Authors: The two mechanisms arise from the power loss density, which can be derived based on Poynting's theorem and non-reciprocity theorem, as we have clarified in our revised paper. The two enhancement mechanisms are related to the two terms in the power loss density, one is the loss in permittivity, the other is the loss in chirality. The first one is related to local field enhancement, and it has been widely used for refractive index sensing with plasmonics (Kabashin *et al*, *Nature Materials* 8, 867 (2009); Im *et al*. *Nature Biotechnology* 32, 490 (2014); Cetin *et al*, *Light: Science & Applications* 3, e122, (2014); just to name a few). The second one is related to local optical chirality density enhancement, it is the one that is responsible for our scheme. However, both enhancements exist when using plasmonic chiral metamaterials, while the first one poses more noises than signals for detecting handedness of molecules. This is also related to the reviewer's question on why there is so much of discussion on adding two opposite spectra to get zero. Because this step is essential. Without this step, both mechanisms contribute to the spectra with molecules, however the first mechanism (near field enhancement) results in a spectral shift due to refractive index.

We respectfully disagree with the reviewer that "there is too much explanation about simple things, such as adding two opposite spectra to get zero". In fact, we think the explanation is not enough, since it seems the reviewer didn't notice that contributions from the first enhancement mechanism (near field enhancement) could cause harm to the sensing scheme. To help the readers understand this mechanism better, we have added further explanation in the Supplemental Information to make our statement clearer. Again we want to emphasize that adding the two opposite spectra to get zero is not trivial for the physical reason that this important step removes the large background signals from the metamaterial itself and cancels out possible contributions from spectral shift due to refractive index changes (please refer to the Supplemental Information and derivations therein). Therefore, the extensive discussions on page 5-6 of the manuscript aim to point out why only one of these mechanisms is important to our design, namely the near field chiral enhancement, rather than both.

Reviewer: It is not clear why optical chirality would not also enhance the CD of the meta-molecules.

Authors: In principle, the chiral molecules may also enhance the CD response of the metamaterial inclusions. However, this effect is not relevant for our sensing platform, since we calibrate the metamaterial sample. Furthermore, we like to point out that the additional interaction coming from the molecules on the metamaterial platform is very small, given the limited quantity of molecules.

Reviewer: For clarity, the authors should use well established notations for the chirality parameter, optical chirality and dissymmetry factor. The physical processes of their interactions should be clearly stated.

Authors: We have tried to use established notations as much as we could, however, here many derivations are introduced for the first time, and these parameters have important physical meaning. In fact, in our manuscript the chiral enhancement factor is directly related to the well-established notation of optical chirality, and optical chirality is directly proportional to the dissymmetry factor. In the revised manuscript, we added this connection per the reviewer's suggestion.

Reviewer: The figures are mostly understandable and are of good quality. Some minor remarks:

- Figure 1 is fine

- Figure 2, the dimensions of the nanostructures should be specified. Having a 500 nm scale bar is not helpful.

Authors: In our original submission, we have already included the dimensions of the nanostructures in the Methods section, Experimental details. For reviewer's reference, we quote our words here "The dimensions of the unit cell nanodipole is $220\text{ nm} \times 50\text{ nm} \times 40\text{ nm}$ (length \times width \times thickness). The unit cell is embedded in a square lattice of $300\text{ nm} \times 300\text{ nm}$." We didn't put these dimensions in the main text for the reason of conciseness.

Following the reviewer's suggestion, we also added these dimensions to the caption of Figure 2 in the revised manuscript.

- Figure 3, it took me a long time to understand that the left side is experiment and the right is theory. This should be clarified on the figures themselves. The CD spectrum of the molecules should be provided.

Authors: In our original submission, we have this information in the figure caption. Per the reviewer's suggestion, we also added "Experiment" and "Simulation" on top of each panel of figure 3 in the revised manuscript. In addition, per the reviewer's suggestion, we added the CD spectra of the molecules in the Supplemental Information, section 1.6.

- Figure 4, the CD spectrum for the molecules should be provided.

Authors: Per the reviewer's suggestion, we added the CD spectrum for both molecules of Figure 4 in the Supplemental Information. These spectra are from literatures.

Reviewer: The data do not support the claims very well.

- The difference in the experimental spectra in figure 3 is not very large. How reproducible is this? The authors say that it is in the text but they do not show any evidence in the manuscript or in the supporting information.

Authors: The reviewer criticized the small difference in the experimental spectra in figure 3. If the reviewer refers to the small spectral difference in panels b and c, we thank the reviewer for raising up the **insignificant shift** in the spectra. This is exactly the point we are trying to make in the paper, and this is why we proposed this new sensing mechanism, which is NOT based on detecting the spectral shift. Please note that with such an insignificant spectral shift, with our method, an obvious difference in the *sign* of the resulting sumCD spectrum is detectable, which is shown in figure 3d.

The reproducibility is shown from the standard deviation of our measurements. We modified the sentence regarding multiple measurements to remind the readers about our statistics, it is now reads "Even with our expected fabrication imperfections, for all measurements reported in the paper, we have been able to consistently reproduce the molecular chirality measurements several times using different samples, evidenced by the standard deviations in the figures."

Reviewer: The authors should measure more molecules and give evidence that their method works. Kadodwala's group investigated a whole range of molecules.

Authors: We have carefully chosen the range of molecules based on molecular weight in our experiments. For example, in figure 3 we demonstrated the sensing mechanism on 1, 2-propanediol, which has a molecular weight of 76.09g/mol (or 76.09Da). To date, the CD measurements of these molecules are limited by either large quantity (conventional CD spectroscopy), or through a novel measurement but involves quantum computation (Patterson et al, Nature 497, 2013). In Figure 4, we demonstrated our mechanism also works for larger molecules such as Concanavalin A and Irinotecan Hydrochloride, which have molecular weights of 104kDa and 623.14Da, respectively. These examples covers a larger range of molecules than what have been demonstrated in Kadodwala's group, who studied mainly proteins with molecular weight in the kDa range.

Reviewer: This work claims the use of a flow cell, but it is not clear whether the molecules were flown through the cell while measured or whether the authors just filled the cell up and then measured.

Authors: We thank the reviewer for raising this question. The cell is first filled, then the measurement is performed. As we mentioned in the text that "The S enantiomers were first flown into the flow cell, and we collected their corresponding CD spectra on +60° metamaterials and -60° twisted metamaterials respectively,....."

To avoid confusion, we modified the text into "The S enantiomers were first *filled* into the flow cell, and we collected their corresponding CD spectra on +60° metamaterials and -60° twisted metamaterials respectively,....."

Reviewer: In figure 4, the authors claim that the CD changes because this is typical of large molecules with several chiral centres. This seems rather naïve. In principle, the CD response can be accurately calculated for simple molecules such as amino acids. When the molecules have more than one chiral centre it becomes much more difficult. For example, the CD spectrum of proteins was never completely understood. Yes, CD changes according to resonances near chiral centres but often the resonances overlap for all centres. A general rule is that (at least for simple molecules) the CD effect changes sign (bisignate) if it is due to exciton coupling (through space coupling between 2 chromophores).

Authors: We thank the reviewer for pointing out our sentence in the caption of Figure 4. Our original sentence is "For a large protein with different chiral centers, the CD signal is expected to show spectral oscillations with changing signs". This is not our claim nor the focus of our study, it is simply a background sentence that we derived from the literature: e.g. Yamada *et al*, J. Org. Chem., 75, 4146 (2010). We agree with the reviewer that for simple molecules, exciton coupling can explain some of the bisignate CD spectrum. However for some molecules, bisignate CD spectrum was also explained as conformational equilibrium and solvational equilibrium (book by David Lightner, Jerome Gurst, *Organic conformational analysis and stereochemistry from circular dichroism spectroscopy*). In the same book, it also mentioned that for molecules such as (-)-menthone, "the origin of bisignate Cotton effect comes from the presence of at least two conformations that have oppositely-signed Cotton effect". On the other hand, for some molecules

chiral centers can be related to Cotton effect, a more recent literature from Yamada *et al*, (Mar Drugs. 14, 74 (2016)) shows a direct relationship between Cotton effect and absolute configuration in Pseurotins.

In the revised manuscript, we modified the sentence in Figure 4 caption into “For a large protein with different chiral centers, the CD signal is *more complicated*, and *may show* spectral oscillations with changing signs”.

Reviewer: Here, the authors do not show the CD of the molecules whereas, in Figure 2, the CD of the nanostructures does show a sign change depending of wavelength.

Authors: Per the reviewer’s suggestion, we included the CD spectra of all the relevant molecules in our revised supplemental information, section 1.6.

Regarding the sign change in the CD of the nanostructure, we have to emphasize that this CD spectrum is our background signal, and it doesn’t contribute to the sign of the final sumCD. We want to remind the reviewer that any of the oscillating features in the CD of the nanostructures have been cancelled out by summing up the spectra. Again the summation is a very important step to remove background signals, as pointed out by Reviewer 2.

REVIEWERS' COMMENTS:

Reviewer #1 (Remarks to the Author):

This version of the manuscript is far clearer than the previous one I reviewed. Its nice to see the modelling of optical chirality. I notice the Kadowala group as a recent paper out (in the last month or two) in Nano Letts which probably should also be referenced. The manuscript could now be published,

However, my principal reservation on the level of novelty remains, but the threshold level for novelty I guess is an editorial decision.

Reviewer #3 (Remarks to the Author):

In this revised version the authors have made efforts to address all the numerous comments from the Reviewers. Overall the quality of the manuscript is improved. The main issue with this work, as highlighted by both Reviewer 1 and 3, is novelty. The authors argue that their work is not identical to previous publications, which is certainly true but the manuscript still does not meet the criteria for novelty and significant scientific advance that would justify publication. The text would be better suited to ACS Photonics or Advanced Optical Materials.

Response to the Reviewers' Comments on the Manuscript NCOMMS-16-01155B-Z entitled "Chirality Detection of Enantiomers Using Twisted Optical Metamaterials" submitted to Nature Communications

We would like to thank the reviewers for the careful reading of our revised paper. We are encouraged by both reviewers' overall positive remarks. In the following, we include a point-by-point response to the comments of each reviewers. We are also attaching a revised manuscript that includes updated text to address the residual reviewers' concerns.

Reviewer #1:

This version of the manuscript is far clearer than the previous one I reviewed. Its nice to see the modelling of optical chirality. I notice the Kadowala group as a recent paper out (in the last month or two) in Nano Letts which probably should also be referenced. The manuscript could now be published.

However, my principal reservation on the level of novelty remains, but the threshold level for novelty I guess is an editorial decision.

Authors: We thank the reviewer for acknowledging the improvement of our revised manuscript. Per the reviewer's suggestion, we added the reference to Kadodwala's paper, Ref. 15 of the revised manuscript.

We take the opportunity to stress and clarify again the novelty of our work. In all existing publications, the sensing mechanism has intrinsic limitations in achieving high sensitivities. Our work is innovative, especially in the following three aspects: 1) our mechanism **doesn't rely on spectral shift** to detect the handedness of the molecules, but on an absolute sign difference in the \sum CD signal. This aspect is important to achieve high sensitivities. 2) Our multilayer chiral structure can amplify chiral-chiral interactions between the metamaterial and the molecules, and translate this amplification to the far-field (10 deg vs 0.1 deg); 3) our approach provides large flexibility to easily tune the enhanced circular dichroism to a desired frequency. As a result, we achieve zeptomole chiral detection, which has not been reported to date.

We have highlighted all these aspects of novelty in the Introduction section of the final manuscript.

Reviewer #3:

In this revised version the authors have made efforts to address all the numerous comments from the Reviewers. Overall the quality of the manuscript is improved. The main issue with this work, as highlighted by both Reviewer 1 and 3, is novelty. The authors argue that their work is not identical to previous publications, which is certainly true but the manuscript still does not meet the criteria for novelty and significant scientific advance that would justify publication. The text would be better suited to ACS Photonics or Advanced Optical Materials.

Authors: We thank the reviewer for acknowledging the improvement of our manuscript. We have stressed in the previous response and in the revised paper the elements of novelty and impact in our work. In addition, in the Discussion section we added more comments on how to further improve our sensing mechanism.